# The Social and Reproductive Challenges Faced by Free-Roaming Horse (*Equus caballus*) Stallions

**DOI:** 10.3390/ani13071151

**Published:** 2023-03-24

**Authors:** Aleksandra Górecka-Bruzda, Joanna Jaworska, Christina R. Stanley

**Affiliations:** 1Department of Animal Behaviour and Welfare, Institute of Genetics and Animal Biotechnology, Polish Academy of Sciences, 05-552 Magdalenka, Poland; 2Institute of Animal Reproduction and Food Research, Polish Academy of Sciences, Department of Gamete and Embryo Biology, 10-243 Olsztyn, Poland; 3Animal Behaviour & Welfare Research Group, Department of Biological Sciences, University of Chester, Chester CH1 4B, UK

**Keywords:** stallions, welfare, behaviour, reproduction, free-roaming, social behaviour

## Abstract

**Simple Summary:**

The stallion is a horse like any other horse, and social interactions are an important part of daily life. Although many free-roaming stallions fulfil their life’s purpose, reproduction, only those with special characteristics enabling the formation and long tenure of a harem can sire a large number of offspring. The harem leader role is, however, a difficult one; the welfare of free-living stallions can be challenged by a range of factors. In this review, we discuss these challenges and explain how understanding the pressures that have shaped the evolution of stallion behaviour can be used to argue for the importance of the social environment to the stallion in captivity.

**Abstract:**

In captivity, intact male horses, due to their sexual drive, are usually socially isolated from other horses. This lifestyle strongly contrasts with that experienced by horses living in free-roaming, feral, or semi-feral conditions, where adult stallions have several roles in their social group, with successful reproduction being their primary drive. Reproductive skew in wild populations is high; many stallions will fail to reproduce at all, while others achieve high levels of reproductive success, siring a large number of foals. Successful stallions are those with particular characteristics and abilities that facilitate harem formation and tenure, allowing them to successfully take over a harem or establish a new one, protect mares from rival stallions, employ appropriate social behaviour to maintain group cohesion, and avoid kin-mating, for example through kin recognition mechanisms. Whilst the life of free-living stallions is far from stress-free, they retain ancestral adaptations to selection pressures (such as predation and competition) exhibited by their natural environment over thousands of years. Here, we discuss the challenges faced by free-living horse stallions, the roles they play in social groups, and their resulting social needs. By understanding these pressures and how stallions react to them, we highlighted the importance of the social environment for the stallion. It is hoped that a better understanding of wild stallions’ lives will lead to their needs being more clearly met in captivity, reducing stereotypical behaviour and improving welfare.

## 1. Introduction

Domestic horses (*Equus caballus*) are exceptional amongst farm animals in terms of a specific characteristic that is favoured by humans: workability. Both males and females are used in human activities. The majority of males, when not selected for reproduction, are granted long lives as geldings (castrated male horses) in captivity. Geldings, showing lower excitability [1], are much easier to handle and house with females (mares). They can therefore enjoy a social life, supporting high levels of welfare in this social animal [2,3,4,5].

In contrast, intact males (stallions), due to their sexual drive, are usually socially isolated from other horses in captivity [6,7]. Since horses are social animals, such social isolation of stallions frequently results in behavioural disorders, stereotypies, or aberrant behaviour during mating/semen collection [8,9] and even reduced fertility [7]. For a large part of their lives, domestic stallions are used for breeding purposes. In natural conditions, copulation results from courting behaviour, which usually occurs between horses that remain together beyond the breeding season. Reproductive activity ends very quickly when the mares become pregnant. This is in clear contrast to the management of domestic stallions, which have intensive and long periods of sexual activity, usually not including any social interactions with mares and other horses, such as foals and immature individuals of both sexes. The decreased welfare of domestic stallions used for reproduction was exhaustively described in [6,7,8,9] and will not be a focus of this review. Instead, we focus on stallions’ experiences in semi-natural conditions.

In free-living conditions, males may also be challenged by several factors, including occasional starvation, social instability, or aggression from other stallions resulting in combat leading to wounds and fatal incidents. These aspects of feral stallions’ lives and reproduction are only rarely addressed in the scientific literature. There is much debate as to the extent to which the welfare of free-living animals should be of concern to society; most agree that where human impacts have negatively affected animals’ lives, interventions should take place, but the argument is less clear in the case of natural pressures, such as inclement weather and injuries resulting from competition [10,11]. We argue that it is essential that welfare threats to free-roaming stallions are understood. Interventions are not always necessary, as such challenges are, in many cases, resolved naturally. However, in some cases (such as where horse populations’ ranges have been artificially reduced by fencing), changes to management or removals might be required where poor welfare persists.

In this work, we present a review of the research on the role that free-roaming adult stallions play in the social and reproductive lives of their harem mares and offspring in free-living conditions, as well as potential challenges to their welfare. Among many definitions of animal welfare [12], we use the one proposed by Duncan [13]: ‘welfare (…) corresponds to the absence of negative subjective emotional states, usually called “suffering” and probably to the presence of positive subjective emotional states, usually called “pleasure”’. We discuss these challenges and explain how understanding the pressures that have shaped the evolution of stallion behaviour can be used to argue for the importance of the stallion’s social environment.

### Literature Search

We used literature on feral horses, but also on semi-feral horses (i.e., free-roaming animals owned by individual owners), and, when pertinent, we referred to research on wild species (e.g., Przewalski’s horses, *Equus ferus przewalskii*, and plains zebras, *Equus quagga*), observed both in zoos and in the wild. We conducted a literature search in Google Scholar using the keywords “stallion” and “feral”, “social”, “equine bands/harem”, “behaviour”, “aggression”, “play”, “grooming,” and “cortisol,” and a combination of these terms. The literature cited in previous studies and further references were retrieved using the snowball method. Finally, 172 references were kept.

## 2. Feralisation

It is estimated that horses were domesticated around 6000 BP [14]. Domestication, as proposed by Diamond [15], involves six critical conditions, all of which are fulfilled by horses: the acceptance of a human-provided diet and restrictive living conditions; the ability to reproduce in captivity with a relatively quick reproductive rate; a follow-the-leader dominance hierarchy; and non-aggressive behaviour towards humans. Within these requirements, two reproductive characteristics are considered: the possibility to reproduce in human-created conditions and a reasonably high reproductive rate. The activity of humans related to the protection of their herds of food-producing herbivores led to the extinction of large predators in many areas. This activity, combined with the introduction of horses to regions devoid of big predators (e.g., Australia), resulted in a high rate of survival for escaped horses.

Feral horse populations persist worldwide; although the direct ancestors of domestic horses are extinct, the versatile potential of horses to manage and reproduce well both in captivity and in the wild [16] has frequently led to their feralisation. While some breeds of domestic horses are reproductively handicapped (e.g., the Thoroughbred horses [17,18,19]), the reproduction of horses in feral conditions is so effective that it often poses the reverse problem, the overproduction of foals. Even in pasture breeding, when mares are kept in free-moving groups with a stallion during the reproductive season, reproductive performance was found to be 10% higher than when using “in-hand” breeding in Hucul and Islandic horses [20,21]. This evidences that in an environment devoid of human management, equine reproduction is more efficient. Observations of feral herds can give significant insight into the socio-reproductive behaviour of horses and therefore help us understand why socially isolated stallions’ welfare is disturbed in captivity [6,7,8,9].

## 3. The Social and Reproductive Organisation of Equids

Thanks to a large volume of observational data for feral groups of horses [22,23,24,25,26,27], wild zebras [28,29], and Przewalski horses [29,30,31,32], the social and reproductive organisation of most extant equid species is well understood. Relatively less is known about other equid species such as kulans (*Equus hemionus*) and kiangs (*Equus kiang*). In equids, social, territorial, and reproductive organisations are interconnected. All equids are polygynous species [28], but two socio-reproductive systems can be differentiated. These are female defence polygyny, a system based on familial groups where stallions defend their mares but not a territory, and resource defence polygyny, a non-familial system where stallions defend a territory and mate with mares entering this area. A key difference is that where female defence polygyny is in operation, mares remain with one (or more) stallions throughout the year, whereas with resource defence polygyny, the sexes tend to segregate outside of the mating season. The system adopted is predicted by the environment: in areas where resources are relatively widely distributed and therefore cannot be defended, female defence is the only option, whereas where resources are clumped together, they can be defended, and so resource-defence polygyny occurs. Although domestic horses usually show female-defence polygyny, Gersick and Rubenstein [33] showed that where water and feeding resources are limited, feral horse stallions are able to defend resources, and mares therefore move independently across the area, resulting in resource-defence polygyny. This flexibility has allowed equid species to persist worldwide across a vast range of habitats, from the waterlogged marshes of the Camargue [34] to the arid deserts of Australia [35].

Most feral horses live in familial groups (also called harems or bands; see later definitions), which comprise the reproductive unit of the adult male(s) and female(s) that live together all year round, rearing generations of their offspring until they become independent. Young males and males who are unable to lead a harem of their own live in bachelor groups; this can be an important stage in a young male’s life in terms of the development of social skills and allow males to develop social bonds with each other and maintain a hierarchy [32,36,37]. In addition to this basic group structure, horses can form multilevel societies (herds). These are composed of both familial groups and bachelor bands [38] and can show social facilitation and flight synchronisation in times of danger [5].

According to a new classification of uni-male multi-female units [39], it could be proposed that domestic horses (*Equus caballus*), Przewalski’s horses, mountain zebra (*Equus zebra zebra*), and plains zebra form true harems, whilst Grevy’s zebra (*Equus zebra Grevyi*), domestic (*Equus asinus*), and feral asses (kulans *Equus hemionus* and kiangs *Equus kiang*) form fission-fusion “coercive consortships” as a result of the latter species showing mostly resource-defence polygyny. This proposed classification considers the types of relationships between and within sexes. According to this classification, territorial stallions are intolerant of satellite males and have short-term female tenure during the breeding season, while the females, although they may form small groups beyond the reproductive season, are less closely bonded than are females that live in year-round harems. In non-territorial equids such as domestic horses, as a general rule, males are also intolerant of satellite males; however, multi-stallion groups have been observed [38,40,41]. For this reason, we use the terms “harem” for a familial group including one male and “band” for a familial group with more than one adult male or when the given paper does not mention the exact number of males in the familial group.

In horses, social bonds are an extremely important part of daily life. The bond between mares and the stallion(s) and male-female interactions are usually consensual, and, importantly, the males have a relatively long tenure as the band stallion(s). Tenures of over ten years have been reported in the Assateague Island [42] and Camargue [34] populations. Group membership is relatively stable; mares may stay together for most of their lives, allowing these strong bonds to develop, and they may even stay together after the death of the harem stallion [23,43]. Band/harem members can split off to graze elsewhere [44], but they will come together at a later point or when the stallion rounds them up. Horses will mostly rest, socialise and graze together as a group, and the stallion will remain with the mares independently of their status: whether pregnant, with a foal at foot, in oestrus, or barren. Bonding is crucial for all domestic horses [4,5,23,45], including stallions, and when free-living, they are rarely solitary.

## 4. A Male Foal

Male foals (colts) differ in social behaviour from their female counterparts (fillies) from a few weeks of age. Male foals play more than fillies, play more with each other, and initiate playing bouts more often [27,46,47]. They play more with the sire than with other adult mares other than the mother, which often chases them away [27,46,47,48]. This is an important result since the father’s presence clearly regulates the group’s social (affiliative and agonistic) behaviour [49]. Specific patterns of colts’ play involve rearing, nibbling, mutual chasing, faking kicks, or neck-fighting [47,50]. As proposed by many authors [27,47], such play elements serve later as fighting skills during real conflicts between adult males. Adult stallions rarely play with each other [31].

Both male and female juveniles stay in the familial group after being weaned by the mare, usually up to around two years old. Although male foals appear to receive more affiliative behaviour and investment from their mothers [51,52], artificially weaned colts seem to be less dependent on their mothers than do fillies a few months after this separation [52]. However, as weaned yearlings, they still approach the dam when they have this possibility, suggesting that at this age, they still need the social support of their mothers [52].

### Challenges for Male Foals

Both male and female foals are more prone to fatal incidents than adult horses due to their smaller size and higher vulnerability. Because their spatial orientation and maternal recognition abilities are very low in the first hours after birth, they easily lose contact with the dam. There have also been reports of infanticide in domestic [53,54,55] and Przewalski’s [56,57,58,59] horses. Despite there being only one recorded case of infanticide in feral horses [53] (except for in newly formed groups of horses released to the wild [55]), the probability of infanticide cannot be completely excluded (discussed in more detail in Section 10: Kin-mating, infanticide, and feticide).

Foals can also be more at risk for predation; foals of domestic horses have been recorded as being preyed on by wolves and cougars more frequently than adults [60,61,62,63]. Although predators were not able to effectively reduce the population of feral horses in North America [62], a significant reduction in the number of free-living Garrana horses and reintroduced Przewalski horses due to wolf predation was observed in Europe and Mongolia [60,61,63]. Juveniles up to one year old may therefore be regular targets of attacks from large predators, and as a result, they may be at higher risk of severe injuries and deaths.

## 5. The Start of Independent Life: The Bachelors

Young males disperse from their natal bands at around two years old (range: 1–5 years) [34,64,65,66]. Whilst Tyler [27] proposed that males under five years old are not perceived by harem stallions as rivals, Stanley and Shultz [67] found male sub-adults received increasingly more aggression with age from other group members than did fillies. They then received more affiliative (supportive) behaviours from their dams, perhaps enabling them to remain in their natal groups for longer and thus benefit from social learning [68], with fillies commonly dispersing a few months earlier [69]. Colts can be expelled from the harem by their sire [23,70,71] or another harem stallion [66,70,71,72] or disperse voluntarily. Sometimes, half-brothers disperse around the same time [69]. Occasionally, sub-adult males do not disperse; two non-dispersed colts have been observed to take over the mares from their father’s harem [68].

The young, dispersed colts often form a single-sex group, called a “bachelor band”. Older stallions who have lost their mares can also join bachelors [23]. Such groups are socially unstable since bachelors leave them to attempt to get their own mares, either by finding wandering, dispersed females or by forming new groups with mares from pre-existing harems. Shortly before becoming harem stallions, young males might leave the bachelor band and become solitary [73]. This is one of two possible situations when a stallion has been observed alone. The second one is the situation of the defeated male, who has lost his harem to another male and therefore has a period of isolation; however, when possible, these males will join a bachelor band [23], Figure 1. Similar adherence to a bachelor band was observed in elderly or sick plains zebra stallions [29]. Although its composition is frequently variable, the bachelor group establishes its own dominance rank and mutual bonds [2,3,74,75]. In Przewalski’s horses, the most dominant bachelors tended to form their own harems when 4–5 years old [21]. A similar age of harem formation by young stallions of domestic horses was found by [72,73,76], but in Berger’s study [66], only bachelors over six years old could win mares by fighting with the resident stallion. This indicates that it is a combination of a male’s age, experience, and specific characteristics that promote dominance that make him a successful harem stallion.

### Challenges for Bachelors

Bachelor bands are unstable due to a high turnover of their members [23,32,36]. This provokes conflicts regarding dominance rank, but when established, the frequency of conflict behaviour decreases [36]. It has been established that testosterone levels are regulated by the reproductive status of the male; a higher level was observed in harem stallions compared with bachelor stallions [36,77]. This result was not confirmed by [78], but this study did find higher faecal metabolite concentrations of oestradiol and epi-androstenediol, which are metabolites of testosterone, in harem stallions when compared with bachelors. No differences in ACTH levels and hair cortisol, hormones commonly related to acute and chronic stress responses, were reported between harem and bachelor stallions [79,80], suggesting that bachelor status is not correlated with a higher stress load in stallions. Since bachelor groups enable social living for defeated or elderly stallions, this unimale formation evidently satisfies social needs for reproductively non-active horse males. It was confirmed that in bachelor bands of juvenile and adult non-breeding stallions, aggression does not usually translate into real combat but is expressed in a ritualised form of prancing, parallel parades, or playing [36,75,81]. Successful attempts at providing stabled stallions mutual companionship after the reproductive season were reported [81,82,83]. As evidenced in Figure 1, the stallions can harmoniously live without any aggression causing mutual injuries.

## 6. Harem and Pair Selection

The reproductive success of stallions is a function of the number of mares they can successfully mate with [84,85] and the length of time they can hold a harem [85]. Thus, stallions have to attract and defend as many mares as possible for a maximal length of time. Whilst stallions can actively take over a harem, mares can also join a harem of their own volition.

Mate selection by female horses might be possible in migrating fillies when a choice of bachelor or harem stallions is available. Dispersing females usually choose, or are integrated into, harems of adult stallions [23,86], often after spending a short period of time in a number of different groups (CS, personal observation). It has been proposed that certain characteristics such as dominance [22,87,88], aggressive behaviour [89], or attentiveness to harem mares [90] may be selected by mares when deciding to stay with a particular stallion. However, the behaviour of any other mares in the group towards the incoming mare could also play a part in this decision [22,24,27,46].

Mares can stay together for most of their lives [43]; the main reason for separation of individuals is the change of the harem stallion. This can occur with the takeover of the whole group by a new stallion, but stallions can also be joined by just some of the mares in a group. Hypothetically, mares can be directed in their mate choice by the MHC (major histocompatibility complex) type of the stallion, as has been observed in domestic conditions by exposing an individual mare to a set of different unfamiliar stallions [91]. However, this hypothesis has not yet been confirmed in semi-feral horses; one study found this not to be the case [92], but the population in this study was skewed toward females. Kaseda [71] observed that mares are able to choose a stallion. After two years of staying barren, an individual mare was observed to leave the harem taken over by her son and voluntarily join the harem of an unrelated stallion [91]. Similarly, contracepted mares were found to be more likely to change harem stallions [93,94]. This indicates that mares can change harems despite any guarding behaviour of the stallion.

## 7. The Role of the Harem Stallion

After the formation of the harem, the most important function of the stallion is to mate with his mares and prevent mating by outside stallions. Due to high pressure from other males during the mating season, stallions must frequently defend their mares from joining another male. In times of high turnover of contracepted mares, stallions are especially alert [93]. When male squeals were played on a speaker, stallions were found to be more vigilant and more likely to approach the loudspeaker compared to when a control sound was played [93]. The stallion is often the individual that is most vigilant when other members of the group are resting or sleeping [27,95]. Stallions therefore spend more time in vigilance and locomotion and less time foraging than do mares [96]. This can have a significant impact on body condition, especially during the mating season.

There are a range of characteristics that can make stallions more successful in terms of harem tenure. Fighting ability can be important; stallions often fight for mares to assure access to fertile females (Figure 2). Due to the presence of competitors, horse stallions are frequently unable to hold more than several mares in their harems at any one time, although much larger harems have been recorded in certain populations (e.g., [97,98]). It seems that the dominance status of the male and certain individual characteristics can also predict harem tenure. Although the weight of a stallion can predict reproductive success [99], neither the male’s size, which would be advantageous when fighting [100], nor his testosterone levels [78] were found to predict reproductive success. Although it has not been experimentally confirmed, some researchers have proposed that stallions’ individual characteristics in terms of their personality are crucial for successful harem tenure [38,86,101]. Experience in the bachelor group can also predict reproductive success; more dominant bachelors can form their harems at a younger age [23,57] and have higher reproductive success [32]. The dominance rank of the mother was found to be predictive of reproductive success in Camargue horses [99].

Another major part of the stallion’s role is group cohesion; stallions will actively search for lost mares and return these to the group. This behaviour is typical of non-territorial species of equids; a plains zebra stallion was once observed taking a sedated mare back to the group by grabbing the fold of her skin with his teeth [28]. Horses recognise individual group members and seem to be aware of the completeness of the familial group [102]. Separated family members (stallions, mares, and foals) have been observed searching for each other when released from transient separation in a field study [28].

The guarding drive is very strong in stallions [49]. Although stallions maintain group cohesion through agonistic herding behaviour, studies have reported that the stallion is not always the most dominant animal in the group [101,103]. His position in the group dominance hierarchy depends on the method and context of dominance assessment. When animals were observed in the context of access to a limited resource (water), stallions usually ranked the highest [22]. However, in an artificially formed pasture-breeding group, when resources were not limited, the stallion was the second-lowest-ranking horse [101]. Except for the case of protecting mares from other stallions’ harassment, stallions rarely initiate displacements from the group [104], do not accelerate the joining process of new members of the harem, and usually take the rear or lateral position in a moving group [105]. In the latter study, when the herding stallion was experimentally separated from his group, collective movement was slower, and the dispersion of the mares was higher as compared to when the stallion was present [105].

### Challenges to Stallions

The most likely welfare issues in free-living stallions are physical injuries from combat [66,106,107] (Figure 2). The injury to the mouth (a piece of lip was bitten off during the fight), preventing the animal from feeding, was recorded in Konik stallion [107]. However, deaths resulting from combat are rarely reported. In Konik horses, some defeated stallions have been found to die soon after losing their harems [108], but decreasing physical condition due to advanced age or illness was probably the main cause of both harem loss and the death. The fate of defeated stallions depends on their age, health, body condition, and the availability of a bachelor group to join. Younger and healthier stallions can remain solitary until they find a disperced mare(s), a male companion, or the bachelor band (Figure 1).

The mating season (Figure 3) itself can be a stressful period for stallions. Endocrinological studies in stabled stallions have shown that blood cortisol levels were increased in stallions in sexual stimulation experiments (exposure to a mare in oestrous with and without the possibility of mating) as compared to basal levels of cortisol in non-stimulated individuals [109,110]. Salivary cortisol levels were also found to increase in stallions during the breeding season [111]. The authors related the increase in cortisol to sexual excitement [109,110] or to the close proximity of other males when housed [111]. It seems, therefore, that increased levels of cortisol in breeding stallions could be a result of frustration due to an expected reward (mating) rather than an indicator of a negative welfare state.

When contraception is used in feral populations, this can have welfare implications for stallions. The use of porcine zona pellucida (PZP), which is one of the immunocontraception methods used in mares, can provoke higher levels of vigilance and guarding behaviour in stallions since contracepted mares can change groups more frequently [94]. Being constantly alert can be challenging for a stallion [94]. This could have a significant impact on body condition due to less time being available for foraging and other maintenance behaviours. However, a recent study did not confirm increased levels of fecal cortisol, and thus chronic stress, in stallions who, mostly due to contracepted mares’ infidelity, experienced frequent changes in their groups [80].

## 8. Agonistic and Affiliative Behaviour

Aggressive and affiliative interactions experienced by individuals play an important role in the welfare of horses. Affiliative behaviours can lead to individual benefits, both in the short-term (such as reduced heart rates) and long-term (such as maintaining social bonds important for successful reproduction [43,44]), while aggressive behaviour can lead to injury in the short-term and group instability and therefore reduced reproductive success in the long-term [41]. The harem stallion has a significant role in the social experience of the familial group, especially in terms of aggression. Stallions have been found to be the instigators of most of the aggressive behaviour in the group in some studies [49,101], while they were not the most aggressive group members in others [49]. It is therefore important to look at the context of such behaviour, as previously commented on for dominance rank assessment. “Herding” or “snaking” behaviour (Figure 4) is the most frequent aggressive behaviour of the stallion; this does not necessarily demonstrate aggression or aversion toward a specific individual but is instead part of guarding behaviour and maintaining group cohesion [48,112]. This behaviour was effective in displacing 98% of mares in Berger’s study [22] and was reported not only in the context of protecting mares from harassment by rivals but also when a predator was sighted [28]. It was also reported that herding and general aggression were less frequent when dependent foals were present in the familial group [113]. Interestingly, the presence of the stallion has been found to actually reduce the number of agonistic interactions between group members, even though he did not intervene in the interactions between the mares [49]. Since low aggression levels within the group often translate into higher stability, the presence of the stallion clearly promotes group stability, which has a direct effect on reproductive efficiency [41,44,86].

Stallions also play a part in affiliative interactions. While male foals are more playful and more popular play partners, grooming is absent in the same male pairs when they are four years old [31]. It has been proposed that affiliative behaviour promoting bonding between males may not be as important as it is in females; bachelor bands are transient, meaning long-term bonds are not necessary, and when in a harem, other males are rivals [41]. Stallions are reported to be less popular partners for affiliative behaviour as measured by the number of allogrooming events and the identity of preferred partners [95,114,115], but this may depend on the season. In spring, which corresponds with the reproductive season in horses, allogrooming was more frequent between stallions and mares than between mares, but this difference was not observed in winter [115]. This can also be due to proximity—stallions are often at the edge of the group and so less likely to be engaged in allogrooming.

Stallions are attractive as social partners to mares when they are approached during oestrus [101]. The proceptive and reciprocal behaviour between the mares and the stallion facilitates optimalisation of mating timing. As oestrous receptive behaviour is clearly correlated with approaching ovulation time [15], by a stallion maintaining proximity to a mare during this time, he can make the most of any mating opportunities and guard the female from other potential partners. Frequent matings increase the likelihood of fertilization and, therefore, reproductive success. Studies on feral horses usually report the number of foals produced/reared per mare, but rarely do this for stallions, as genetic testing is required [71]. When it was confirmed by paternity testing, Konik harem stallions’ siring rate was very high (up to 89%, [16]), including all mares in the group, although other factors such as a mare’s age play a part. In feral horses, elderly or infertile mares remain members of the familial group [116]. Higher-ranked mares were observed to be mated as a priority by the stallion when simultaneously in oestrus with low-ranking mares [117]. However, other studies did not confirm this selectivity on the part of the stallion; dominant mares were probably more efficient at monopolizing the stallion’s attention, but ultimately all oestrous mares are bred by the stallion [99].

The presence of the stallion has been suggested to reduce the frequency of allogrooming between mares [49]. Allogrooming is usually interpreted as affiliative and comforting behaviour [118]. Since it has been reported to have a calming effect on heart rate [115], a group showing lower rates of allogrooming might experience less tension between members than one with higher rates of allogrooming, which is required to reduce tension. This hypothesis has been confirmed by the fact that females with stronger affiliative relationships were not less aggressive toward each other [101]. The presence of the stallion may therefore have a calming, stabilizing effect; in groups without stallions, both mare-mare aggression and grooming frequency were higher [46].

Although stallions are not involved in many affiliative interactions with other band members, this does not mean that they do not have preferred partners [114]. Stallions have been reported to stay in the proximity of specific mares [114]. They have been shown to be more tolerant than resident mares when approached by juvenile offspring of both sexes and respond actively to the affiliative and playful behaviour of their foals [27,47,48]. Foals present “snapping” behaviour toward them (Figure 5), which is proposed to reduce potential aggression toward an approaching foal [27] or to calm the submissive individual [119]. In addition, male foals tended to approach their sire more often than female foals [27].

The presence of the sire can have an impact on the behaviour of youngsters, provoking different social behaviour in male and female offspring [120]. Fillies were found to prefer colts as play partners when a stallion was present, while in the absence of the stallion, they played mostly with other fillies. This special role of the stallion in the ontogeny of the foal can probably influence the future psychological properties of his sons and daughters. More affiliative and playful behaviour between bachelors has also been observed in Przewalski’s horses reared in familial groups with the father as compared to domestic horse bachelors reared in broodmare-only groups [3].

The presence of the harem stallion and his specific behaviour toward other group members may affect natal dispersal and the future reproductive life of his offspring. While stallions’ herd and guard their harem mares, they usually do not interfere with the sexual interactions between their daughters (or stepdaughters) and other stallions [43,121]. The aggressive expulsion of the daughters at their first oestrus was, however, often observed in Konik horses [70,107,108]. In addition, when the continuous presence of the male foal in the familial group with his father is interrupted, the future adult male may present disturbed reproductive behaviour. For instance, higher tolerance to non-dispersed daughters was observed in stallions that were released to semi-feral conditions after being removed from their familial group before natural weaning [68]. Moreover, if such stallions were used for “in-hand” sexual activity in human-controlled breeding, they were also more prone to mate with their non-dispersed daughters [68].

## 9. Familial Bands with More Than One Stallion

Although the majority of studies report one-male familial groups (harems) in feral horses, the presence of a subordinate stallion as a member of the familial group (band) has also been observed [34,38,40]. The proposed explanation for bands with more than one stallion was the cooperation (mutualism) between stallions in the familial band [34], but this hypothesis was not confirmed by Linklater and Cameron [40], who suggested mate parasitism and consort hypotheses for having multiple stallions in a band. It should also be noted that the tendency of non-territorial equids to form large herds may be misleading in terms of stallion ownership of the harem when observing such groups. Long-term observations of social groups in Konik horses showed that the distinct families of two stallions formed such an inseparable herd roaming together that it could be mistaken for one band with two stallions (AGB, personal observation). Indeed, subgroups comprised of different individuals were distinguished within larger herds in studies by [38].

Subordinate stallions might have a specific role within a band. As reported by [23,34,95], the second stallion is always subordinate to the band stallion. The band stallion appeared to have an exclusive right to mate with the band mares; however, paternity testing of the foals showed that the band (and the harem) stallions were not the biological fathers of up to one-third of the offspring [71,99,122]. Subordinate males have also been observed to be involved in guarding the mares against foreign stallions’ harassment [34]. It has been proposed that these alliances between adult males therefore help them increase their reproductive success by excluding competitors [34]. However, the role of the subordinate stallion as a helper in band protection has not been fully confirmed [40]. Single stallions had more mares than stallions in multiple-stallions bands [123]. Experimental withdrawal of the subordinate stallion did not provoke any loss of the group’s integrity by the dominant male [124]. While the transient withdrawal of the dominant stallion caused the subordinate male to take leadership of the group, the return of the dominant stallion re-established the previous hierarchy [3]. This is therefore another confirmation of the individual characteristics of males predisposing them to the role of harem/band dominant stallions.

Adult males can tolerate each other in terms of respecting a group’s dominance hierarchy. This relates to both bachelor and familial bands. In groups with an unstable hierarchy between the stallions, the presence of females may lead to increased aggression and physical injuries [6]. Since in feral horses, the social rank of the stallion is usually established by ritualised behavioural encounters [36,50] to avoid spending energy on real combat [100,124] or during time spent in a bachelor group, stallions in multi-male bands can peacefully co-exist in close proximity with mares present. It seems that the subordinate stallion accepts its role at the cost of being reproductively relatively inactive but with the benefit of being socially satisfied and with the potential to sire a small proportion of the group’s foals. There is therefore little likelihood of multi-male bands posing a particularly high welfare risk to males.

## 10. Kin-Mating, Infanticide, and Feticide

The choice of reproductive partner is believed not to be random in the majority of animal species, including horses [92,125,126,127]. The biological cost of semen production as well as acquiring and guarding the reproductively available mares is high for a stallion [128,129]. Inbreeding is one of the factors suggested to decrease the reproductive success of an individual [130], thus mechanisms that allow to recognize related individuals have evolved [131]. Experiments carried out under controlled conditions suggest that stallions may allocate their reproductive effort depending on the mare’s reference to his Major Histocompatibility Complex (MHC) [129]. It is speculated that the testosterone level in the semen is higher in the semen of stallions exposed to MHC-dissimilar mares. The same authors indicate that horses recognize MHC, and stallions may use it as a reference for kin recognition [129]. In light of this, feral and semi-feral stallions may use the MHC of mares to recognize and avoid breeding-related females [91]. Notwithstanding, observations made on feral and semi-feral horses do not support this hypothesis [92]. Harem stallions may associate with a group of mares for multiple breeding seasons (years), and other mechanisms ensuring avoidance of breeding with their own daughters and, in some cases, dams, are observed [68,70,87,121]. In natural conditions, foals learn to recognize members of their group from the time of birth [132]. Associations with harem members, both as a foal and later as an adult, may allow a stallion to memorize and recognize familial horses during later interactions [121]. Observations by Berger and Cunningham [121] as well as our own [70] indicate that familiarity is involved in the mechanism of inbreeding avoidance in feral and semi-feral horses. Both fathers and stepfathers avoid mating with their daughters [70,121]. Hence, not the level of relatedness, for example, MHC similarity or dissimilarity, but familiarity may be an important factor in the inbreeding avoidance mechanism in stallions. In support of that hypothesis, it seems that recognition of familial horses is suggested to be time limited, and mating between a stallion and his daughter was observed when the stallion encountered the mare after 19 months of dispersal [121]. The social environment of the stallion, from the moment of birth, next to familiarity, may be another factor involved in the development of its later appropriate sexual behaviour [8,9,70,133]. Indeed, stallions who experienced a period of captivity during which they bred mares under human-controlled conditions showed incestuous behaviour towards their daughters after being released again in natural conditions [70,131].

Infanticide has been reported to occur in naturally living horse populations. An increase in maternal protection of foals not sired by the current harem stallion was reported [53,134]. The occurrence of infanticide, defined as killing a foal sired by another stallion, and feticide, a mare’s counterstrategy to a stallion’s aggressive behaviour towards foals, have been explained in the literature in terms of them increasing reproductive success [135,136], but there are arguments as to why this may not be the case.

Infanticide is believed to increase the reproductive success of males in multiple species [135]. Killing nursing offspring is known to shorten the time until a female returns to oestrus and becomes reproductively receptive; it is therefore in a male’s interests to kill existing non-kin offspring when he takes over a harem [136]. Pregnancy in horses lasts around 11 months, and the mare starts cycling again within 8–14 days postpartum [137]. If not mated, she remains fertile throughout the breeding season, even when nursing a foal. Therefore, there seems to be little potential benefit to infanticide in horses. The killing of both related and unrelated foals by stallions has been observed, albeit infrequently [53,55,138]. Interestingly, mares did not allow infanticidal stallions to mate with them. Feh and Munkhtuya [138] proposed that stallion infanticide is a pathological behaviour rather than a reproductive strategy. In the Przewalski’s horses, they observed that only stallions that were born in zoos and experienced early separation from their natal herd acted aggressively towards newborn foals [138]. Feticide is believed to be a female’s response to a male’s infanticide in order to reduce the biological cost of pregnancy for the offspring, which may be lost. It has been proposed [139,140,141] that mares may experience pregnancy loss after embryo implantation when they are exposed to a male, either a gelding or a stallion, that did not sire the pregnancy. This can even occur through indirect contact, for example, over the fence of the pasture. According to these authors, mares who are at least six weeks pregnant (after embryo implantation) try to solicit a male with whom they have not been mated. If they are deprived of physical contact, they are at a higher risk of pregnancy loss in comparison to mares who are allowed direct contact with other stallions or geldings (reviewed in [136]). It should be acknowledged that around day 42 of pregnancy, structures called endometrial cups are formed around the site of implantation (reviewed in [142]). Most importantly, once these arise, the endometrial cups exist whether the mare is pregnant or not [143]. Hence, if a mare experiences a pregnancy loss after endometrial cups have formed, she will not return to oestrus for approximately 120 days, the lifespan of endometrial cups [143]. The stallion whose presence may have caused the feticide through its proximity to a pregnant mare will therefore gain no reproductive benefits. Hence, neither infanticide nor feticide are two phenomena that are likely to bring any immediate benefits to a stallion, regardless of their living environment.

## 11. Over-Reproduction

High reproductive efficiency is typical of feral horses [9,144], which can lead to significant welfare issues and conflict with local communities. When horse populations are not under pressure from large predators [61,62] or humans, they may outgrow the available habitat and pose threats to local fauna [145] and flora [146]. Overgrazing is also a problem for the horses themselves, causing decreased body condition, increased parasite infestation, starvation, and potentially fatal consequences [147,148]. A shortage of grazing not only has a direct impact on the body condition of stallions, but it can also mean they are unable to monopolise groups of mares, who become more motivated to search for new feeding locations than they are to remain in their familial group [33].

To avoid the consequences of overgrazing and population overgrowth, several methods of reducing reproduction have been applied to feral horses [149]. The most common method, contraception with porcine zona pellucida (PZP) preparations, was found to be effective in reducing the number of foals in feral horse populations [150,151]. Early studies did not find any effect of contraception on the social interactions between mares and stallions [152]. However, as suggested by Ransom [153], the social composition of groups with contracepted females may be disturbed due to the artificially reduced number of foals [154]. Similarly, culling or the removal of surplus animals can disrupt the social networks of equine groups.

The GonaCon-B contraceptive vaccine had no measurable effect on the social behaviour of horses, but a reduction in the number of individuals due to removals affected stallions’ behaviour: they herded and guarded their harems less when the population was reduced [111,152]. It was also found that PZP-contracepted mares showed changed time budgets [94], lower harem fidelity, and changed social groups more often than did control mares [80,93,94,154]. As stressed by the authors, a change in group composition often provokes conflicts between resident and incoming mares, changes in the relations between the harem stallion and resident mares, and a disruption to group stability, which is fundamental to reproductive performance [45]. Since contracepted mares are regularly cycling and showing oestrous behaviour, they receive more frequent and prolonged sexual attention from the stallion, which generally increases his vigilance and reduces his available foraging time [148]. Such continuous sexual activity does not occur in feral stallions with non-contracepted mares since after the reproductive season, all mares are either pregnant or anoestrous, meaning sexual activity ceases and stallions can regain some of the body condition lost during the mating season.

Other methods of population control are also available. One of these is surgical castration (for geldings). No decrease in maintenance and agonistic behaviour was found, but less marking and more affiliative behaviours were observed [155]. However, fewer gelded stallions were able to hold a harem one year following castration; intact bachelors of the same cohort successfully formed harems. Sneak copulation was proposed to have occurred as there was only a temporary (one season) decrease in the number of foals born [155]. One study showed most vasectomised males remained able to hold their harem over the following two years, although other males were attracted to the groups due to constant mare cycling [156]. These procedures do, however, require heavy sedation or general anaesthesia for a feral stallion, incurring risk and expense. Stallions immobilised in the field can also have difficulty relocating their social group or losing their mares to another stallion during their absence [157]. A method of non-surgical castration of stallions was tested on one of Przewalski’s stallions [158]. The method resulted in a decrease in spermatozoa number, motility, and testosterone production. The stallion therefore lost his position as a harem stallion. Other methods of male contraception, such as a laparoscopic vasectomy [159], the use of the indenopyride derivative RTI-4587-073(l) [160], or chemical castration [161], have also been proposed for stallions. Although the consequences of these treatments on behavioural and social functions of stallions have not been investigated in full, it could be hypothesized that they would be similar to those resulting from surgical castration or vasectomy. The most likely effect would be a change in the stallion’s physiology that would cause changes to the mares’ behaviour; this would be independent of the method of stallion contraception.

There are potential ethical issues surrounding male reproductive control, as discussed in [162]. These include concerns that surgical or chemical treatments can negatively impact the physiological and psychological integrity of these animals. However, it can also be argued that the potential costs of reproductive control are outweighed by the relative benefits to males, who are then able to live a close to natural life in social groups instead of facing an uncertain fate as a surplus feral stallion that must be removed from the population [162,163,164].

## 12. Captive Stallions

In the present review, we focus only on free-roaming stallions, but the knowledge gathered clearly indicates conditions that are optimal for all intact male horses. Stallions kept in human-controlled conditions are usually assured freedom from hunger and thirst and from discomfort around resting; they are sheltered, vaccinated, dewormed, and their hooves are maintained in good condition. When used in breeding, they do not have to fight for access to fertile females. They often have higher levels of sexual activity and produce more foals in total than do free-roaming stallions. It could therefore be argued that they experience a life of relative luxury.

However, in many cases, stallions live a solitary life in a relatively barren environment, often having no direct social contact with other horses (Figure 6). As summarised in the current review, from foalhood until old age, the social life of captive stallions, at least in some important stages of development, strongly differs from that in natural conditions. They are separated from the mothers that they still need [165] much earlier (usually at 4–7 months [166]) than would occur under natural conditions (around 2 years of age). They are devoid of contact with females at all life stages, lacking natural learning about mares’ sexual behaviour [9]. They are asked to mate with any oestrous female, often one they are not familiar with. They can even be used in kin breeding if the breeders decide this is required (e.g., [167,168,169]). Although in most breeding facilities juvenile colts are reared together [2], the majority of them are gelded, and only a few chosen individuals are kept intact. From this time on, adult stallions are socially isolated. The social role they evolved to play is not fulfilled, and the comforting social feedback from their familial or companion groups is not received (Figure 6). As mentioned previously [6,7,8,9], frequently this provokes frustration that results in abnormal sexual behaviour, stereotypies, and self-mutilation.

A number of specific recommendations can be made to safeguard and improve captive stallion welfare; many of these have been outlined in previous publications [6,7,82,83,123]. However, our review gives a baseline to refer to when planning and managing stallion housing facilities; only by understanding the social environment in which stallions have evolved can we hope to replicate this where possible in captivity. Obviously, in-pasture breeding or maintaining a stallion with mares and foals throughout the year would be the most desirable method of satisfying a stallion’s needs [48,49,83,123]. Where this is not practical, managers need to ensure alternative arrangements are in place to support stallions’ natural drives. The most important and feasible enrichments that can be implemented focus on social enhancement; these can involve housing stallions in groups or at least allowing direct contact between stallions, for example in “social boxes” [6], a very promising method for allowing the formation of social bonds and hence improving the social lives of stallions. Since grazing and constant locomotion in open spaces are the predominant activities in an equine time budget, access to environmentally stimulating outdoor spaces and pastures is essential. In natural conditions, reproduction is limited to a short, seasonal period; in captivity, avoiding unnecessary sexual arousal when mating is not going to take place has been suggested as a way of reducing frustration and potential aggression in stallions [170]. If these simple recommendations can be implemented, significant improvements in stallion welfare will quickly be possible.

## 13. Conclusions

The life of a free-living stallion can be stressful. A number of challenges have been identified here that can be experienced by feral stallions. In a recent study on Italian horses [171], hair cortisol was found to be higher in semi-feral horses than in domestic horses living in captivity. This could indicate that higher levels of chronic stress are experienced by free-living horses as compared with stabled horses. The authors explain this in terms of the high predation rate in their study population. On the other hand, behavioural problems, such as stereotypies, excessive aggression, and abnormal behaviour, including self-abusing mutilations, frequently occur in captive-living populations, yet are never reported in feral stallions; this could be due to all their needs being satisfied. Aside from the reproductive season, when harem stallions can experience extremely high levels of stress due to the constant guarding of their cycling females from rival males and bachelor males are under higher pressure to sneak copulations or seize a harem stallion’s mares, feral stallions generally experience a stable social life, either with closely bonded harem mares or with fellow males in a bachelor group. Thus, as aptly stated by Briard et al. [172], the stallion is a horse just like any other. It is only in captivity that we often subject stallions to a life in isolation; this is frequently where welfare is subsequently compromised. Therefore, despite the risk of injury, starvation, and predation experienced by feral stallions, it could be argued that their welfare levels are higher, as they are well adapted to these pressures.

An understanding of how horses live and survive in their natural environment is essential for those who manage horses in captivity. Horses have been under natural selection pressure for thousands of years, but the pressures imposed by domestication are only relatively recent. This is particularly important in the case of stallions, social animals whose correct physiological and psychological functioning are often highly affected by social experiences (e.g., see [70]). Sadly, normal reproductive interactions and behaviour seem surplus to requirements in the era of artificial insemination. Perhaps by allowing more natural behaviour, the welfare of stallions living in captivity can be safeguarded and issues related to low fertility can be overcome [7].

## Figures and Tables

**Figure 1 animals-13-01151-f001:**
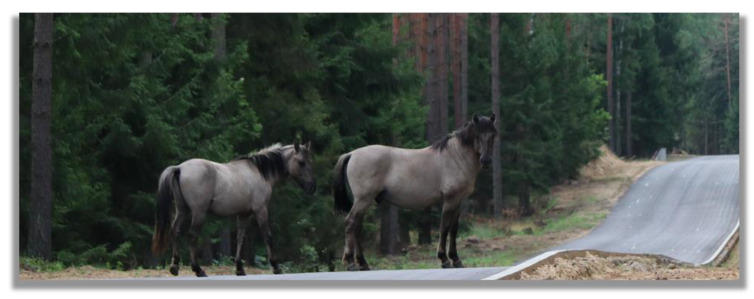
A defeated four-year-old Konik stallion (right) and a recently dispersed two-year-old colt (left) in a bachelor group, Popielno Sanctuary, Poland (photo by Michał Bruzda).

**Figure 2 animals-13-01151-f002:**
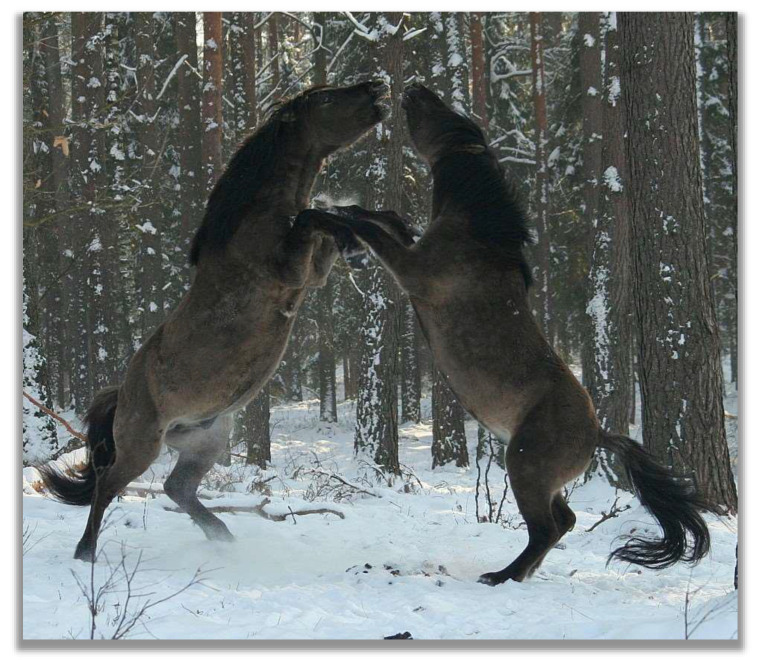
Fighting Konik stallions, Popielno sanctuary, Poland (photo by Michał Bruzda).

**Figure 3 animals-13-01151-f003:**
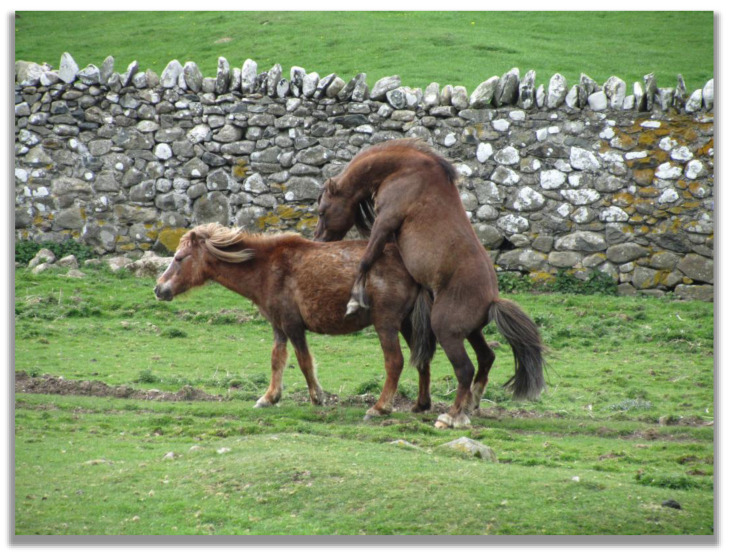
Mating in Carneddau ponies, Carneddau mountain range, North Wales, UK (photo by Christina R. Stanley).

**Figure 4 animals-13-01151-f004:**
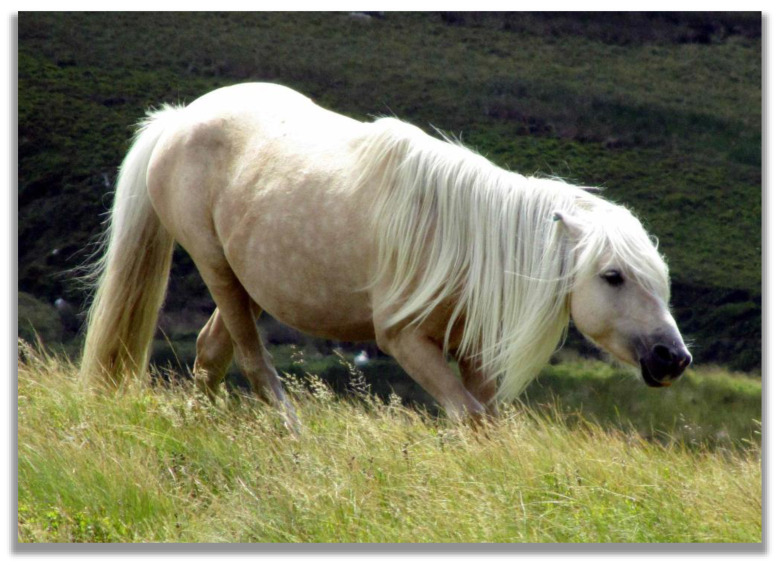
Herding behaviour shown by a Carneddau pony stallion, Carneddau mountain range, North Wales, UK (photo by Christina R. Stanley).

**Figure 5 animals-13-01151-f005:**
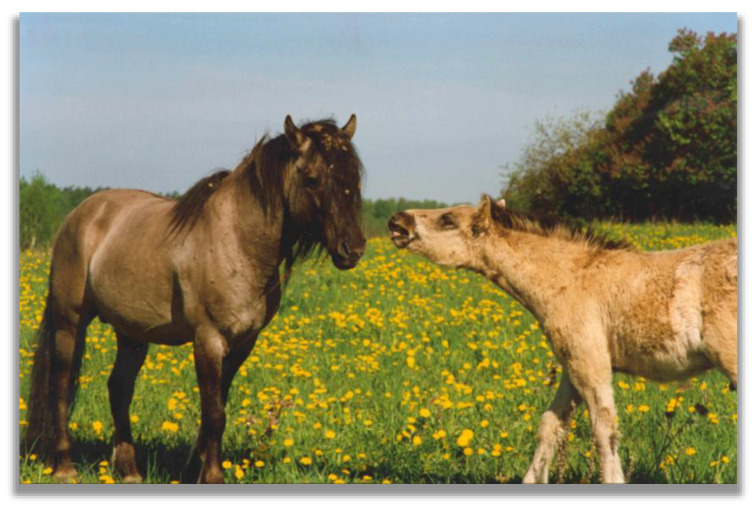
“Snapping” foal approaching a Konik stallion, Zielony Ostrów sanctuary, Poland (photo by Tadeusz Jezierski).

**Figure 6 animals-13-01151-f006:**
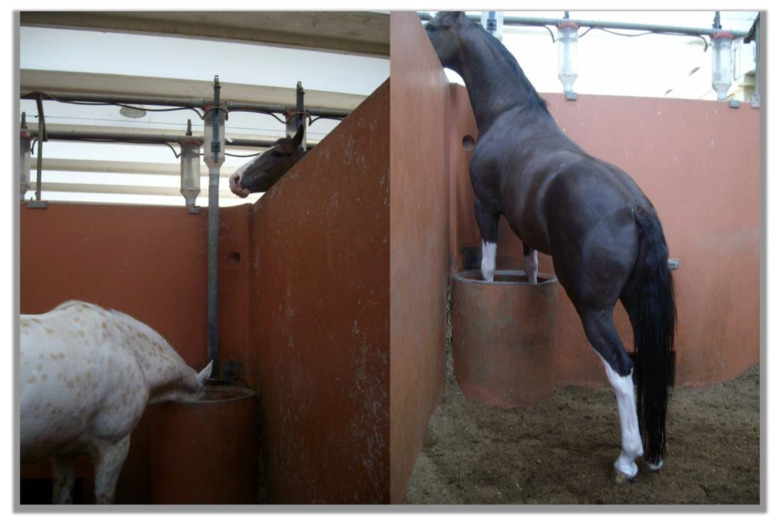
In a barren environment, a stallion (right) searches for social contact with the stallion in a neighbouring box (photo by Aleksandra Górecka-Bruzda).

## Data Availability

No new data were created or analyzed in this study. Data sharing is not applicable to this article.

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
