# Peer review of "The Social and Reproductive Challenges Faced by Free-Roaming Horse (Equus caballus) Stallions"

_animals, 2023, doi:10.3390/ani13071151_

Round 1
Reviewer 1 Report
Review for animals-2198653: Bachelor and family life: welfare and reproduction of free-2 roaming domestic horse (Equus caballus) stallions
This manuscript reviews what is known about feral stallions’ welfare with the goal of applying that knowledge to the betterment of stallions in captivity. The paper is well-written, but unfortunately falls short of that worthwhile and important goal. It would be wonderful if, in their conclusions, the authors could suggest some realistic practices that horse owners could potentially apply to their captive stallions. I know the feral and captive worlds are very different, but even a short list of things that horse owners could do to improve the welfare of their animals would be most welcome, I’m sure. More specific comments (most all to do with grammatical issues and not substance) follow:
Line 28—change to “…allowing them to successfully take…”
Line 69-70—in what context should threats to feral stallion welfare be assessed and reduced? I’m assuming during management practices or in other instances of human perturbation; but this should be clearly articulated as I think interfering with natural processes in an attempt to “improve animal welfare” is a dangerous proposition and often ends in failure.
Line 119—change Dubenstein to Rubenstein
Line 129—change to “…this can be an important stage…”
Lines 188-190—the authors might want to check out Gray 2009, An infanticide attempt by a
free-roaming feral stallion (Equus caballus) in Biology Letters 5, 23–25 doi:10.1098/rsbl.2008.0571. She reports an attempt of foal infanticide in a feral horse population in the Virginia Mountain Range outside Reno, Nevada.
Lines 219-235—there are also cases (though perhaps none of them have been published) in which bachelors will acquire mares and remain together—I have seen as many as three bachelor males do this. The dominant male(s) will defend the female against the other male(s) but they will stay together. I would be surprised if this hasn’t happened elsewhere, though I suppose it’s likely to be rare.
Line 275—delete the space before MHC
Line 328—does the fact that the male takes the rear in a moving group necessarily mean he is subordinate to the other animals? I would think that puts him in the best posotion to protect his mares—he can see anyone coming towards them and can more easily detect and protect them from anyone approaching from behind.
Line 352—change to “…that increased levels…
Line 355—change to “…the use of porcine zona pellucida (PZP) contraception…” Also, as this is the first mention of PZP, I wonder if some explanation of it would be useful. You give a brief description of it later in the manuscript (in Lines 580-587), but again, as this is the first mention of it, some brief description would be appropriate here.
Lines 359-361—fecal (not hair) cortisol was used to assess stallion stress levels in Jones and Nuñez 2023.
Line 436—change to “…more often than female foals.”
Line 471—change to “…subgroups comprised of…”
Lines 473-488—I wonder if even though the greater benefits hypothesized are not realized, if it’s just physiologically less taxing to defend a band with another male—has anyone ever been able to quantify this? It might just beneficial to reduce energy expenditure.
Line 502—change to “random in the majority…”
Line 511-512—change to “…in light of this…”
Line 516—delete “in feral horses.”
Line 554-555—change to “…loss after embryo implantation…”
Line 556—change to “…which did not sire…” and separate into two sentences: “It has been proposed that mares may experience a pregnancy loss after embryo implantation when they are exposed to a male, either gelding or stallion, which did not sire the pregnancy. This can even occur through indirect contact, for example over a pasture fence.
Line 560—change to “…allowed direct contact…”
Line 563—change to “…exist whether the mare is pregnant or not.”
Line 565—change to “…return to oestrus for approximately 120 days…”
Line 576—change to “…fatal consequences [148,149]. Shortage of grazing…
Line 592-3—change to “…showed changed time budgets…”
Line 601—for feral horse populations living in temperate climates, it is not that “all mares are pregnant”, but that usually, that even non-pregnant mares cease ovulating in the fall and winter months (an adaptive strategy given that their gestation period is approximately 11-12 months meaning that mares give birth around the same time the year after conception).
Line 602—change to “…sexual activity…”
Lines 617-619—I am thinking that the vasectomy and the use of an indenopyride is one method—in that case, change “have” to “has” in that sentence. If they are two separate methods, then change “Another method” to “Other methods”. In addition, even though the consequences of these treatments on behavior and social functions of stallions is not known, the authors may like to add a sentence outlining that the results will likely be similar to those we have seen with other methods. In these cases, it’s fairly clear that the methods themselves are not important—it is the physiological changes to the males that lead to changes in mare loyalty and the stallions’ ability to keep a band.
Line 649—I think something is missing from the phrase “…horses have been natural certain…”
Line 654—change to “… in the wild…”
Again, I think that the addition of specific and practical recommendations to horse owners about how they can improve the welfare of the stallions under their care would be an excellent addition to this manuscript.
Reviewer 2 Report
This is a nicely written review of the literature around male horses and their behavior. The authors mention welfare a few times, but the link between the body of the paper and welfare is tenuous at best. Although I enjoyed reading this manuscript I do not quite see what the point is that the authors are trying to make. If this paper is to educate horse breeders then each section should come back to how welfare of breeding domestic stallions can be improved. The manuscript would be improved by making it more pointed and relevant, or else it needs to be framed simply as a literature review.
Several times the authors say that the 6000 years of domestication do not replace the millennia in which horses evolved, but have any horses ‘evolved’ to deal with pressures of captivity? Especially as the rules of that captivity change over time. Until relatively recently it was a lot more common for stallions to be used as riding horses, and thus their welfare was probably higher (in that they would not have been kept in isolation). The authors also seemed to imply that welfare standards should be applied to feral horses, although these same standards would then mean that these horses would have to be managed as livestock. The introduction makes it sound like feral horses need more welfare support, but the introduction makes it sound like domestic horses should be managed more naturally. A consistent message would be helpful.
My other main comment is that although this manuscript represents a laudable review of the literature, they often present data from domestic or zoo horses and those in relatively small reserves as if it represents the same behavior found in wild free-roaming populations. Although studies of horses in pastures, reserves, and zoos can show us details of horse behavior, these behaviors are often vastly different in terms of frequency or scale to those seen in wild populations. I suggest that the authors make these distinctions very clear.
There were minor wording errors and failure to close parentheses throughout, so I suggest the authors review the manuscript carefully.
L27 – “breed themselves” reads strangely. Change to “reproduce”.
L46 – this may not be true – see King, S. R. B., K. A. Schoenecker, and M. J. Cole. 2022. Effect of adult male sterilization on the behavior and social associations of a feral polygynous ungulate: the horse. Applied Animal Behaviour Science 249:105598. However there is a lack of research in to the effect of gelding on horse behavior and long-term effects.
L55 – in free-roaming horses sneak matings between individuals of different groups also occurs.
L107 – kiangs are not considered endangered.
L119 – correct typo of Rubenstein.
L158 – a citation (or multiple) is needed for the statement that “mares usually stay together for life”. I think that this statement should be changed, as female group changes are in fact incredibly common in horse populations, and juvenile females do not always disperse together to the same group.
L161-162 – surely the definition of a group is that they remain together even if at a short distance? Rephrase this.
L164 – this paragraph has not established why bonding is crucial, just that it exists. Rephrase.
L189 – infanticide has been reported in multiple feral horse populations, as cited by the authors. Therefore rephrase this sentence.
L216 – change ‘migrate’ to ‘disperse’. Migration is an altogether different process. Also you should specify that by ‘half-brothers’ you mean males of the same/similar age that grew up in the same harem.
L222 – rephrase ‘stealing’. This word ignores the fact that female choice has a role in this process.
L251 – and play!
L270-271 – although I am sure you are correct, you may also want to discuss how new members of a group frequently receive more aggression. Yet mares still change groups.
L272-273 – see my comment above. In my experience when there is a new stallion some females stay together and others separate, and some females change groups even if there is no change in stallion. I therefore think it is relatively rare for mares to stay together for their entire lives, and would like to see more than this one citation to back up this statement.
L274 – again, I dislike the word ‘steal’ as I think it only tells the patriarchal side of the story.
L298-299 – what evidence do you have of this? How would a stallion assess whether a female is fertile?
L321-322 – this is a critical point! The low frequency of agonistic behaviors in most wild equid populations makes dominance very difficult to assess, and begs the question as to whether it is even relevant to the individuals.
L355 – correct PZA to PZP, and write porcine zona pellucida in full before first use of the acronym.
L380-382 – the intriguing thing to me about herding behavior is that it can also occur in the absence of either conspecific competitors or predators.
L397 – what about play between stallions?
L404 – correct reference formatting.
L460 – I like the term “familial band” but earlier you define this as a harem.
L491-492 – be more clear about what you mean by “when the hierarchy is not respected”.
L493 – this seems like a massive generalization.
L512 – define ‘semi-feral’.
L539 - Hex, S. B. S. W., M. Mwangi, R. Warungu, and D. Rubenstein. 2022. An observation of attempted infanticide and female–female cooperation in wild plains zebras (Equus quagga). Behaviour Online early. Provides a nice review of infanticide in equids.
L546 – Cameron and Linklater found that females that lost their foals (i.e., were not pregnant and lactating at the same time) had greater foal survival of their next foal. Thus infanticide could benefit the stallion.
L548 – rephrase – it is not clear what you mean.
L585 – it would be more appropriate to cite the numerous papers by Nunez on this topic (as you do later in this section).
L649 – a word seems to be missing from this sentence.
Author Response
Please see the attachement.

Round 2
Reviewer 2 Report
While the authors made some changes in response to my comments on a previous draft there are still a lot of changes required in this manuscript. There should be more re-structuring to demonstrate how wild behavior reflects on domestic management, instead of occasionally referring to natural behavior as a welfare issue. It seemed odd to me to refer to injuries stallions incur by fighting or foals potentially dying if their mother was old and so not producing enough milk was a welfare issue. The authors should maybe define what they mean by welfare – to me, welfare is a result of something that is human-caused.
The authors failed to take into account my comments about horses being artificially selected for thousands of years, and also did not respond to my comment about using papers from domestic or zoo horses satisfactorily. Many citations given are when one paper made the point the authors are trying to present, and are often from older work that was done on a very small sample size. It seems a shame to persist old tropes from old research when we now know more and know better.
A lot of the response to reviewers comments revolved around ‘we want to focus on stallions’, yet there is still a lot of material in this manuscript about mares. This could be reduced.
Ultimately the authors need to focus more on welfare issues of domestic horses by contrasting their plight with feral horses.
L30-32 – rephrase to make it clear that equids in general have adapted. It is contentious whether domestic horses have been subject to natural selection in regards to predation and competition as they have been subject to so much artificial selection. Even ‘wild’ populations have only been that way for relatively few generations (hundreds of years at absolute most; most feral populations are much younger than that).
L33 – clarify that you mean horse stallions.
L50 – what do you mean by ‘mostly engaged’? They do not spend most of their time in copulatory behavior! I think that what you are trying to say is that most domestic stallions are kept for breeding purposes. You do not need to say more as you explain in subsequent sentences.
L65 – why is it essential if it gets taken care of naturally?
L76-80 – this is essentially your methods section. I would preface it with “We conducted a literature search…”.
L133 – you could just say “Males who are unable to lead…” as bachelor groups consist of all ages of males, as well as those pre- and post-harem status.
L156-159 – you could remove this if your focus is purely on males.
L164 – I still object to the word ‘usually’ as I do not believe this to be true. I suggest changing it to ‘may’.
L166-169 – I suggest removing this sentence. I understand what you are describing – individuals of a group can spread quite widely while grazing – but this is still very different from fission-fusion social structure. I do not see the point of this sentence.
L170 – change ‘guard’ to ‘remains with’. Guard is not only anthropomorphic but also hard to quantify, and no citation is provided.
L184 – this is not true. I have frequently seen adult stallions play together, and know that other people have observed this too. The paper you cite states that play was “almost never” observed in adults, suggesting it was observed at least once. I suggest removing this sentence.
L185 – as horses are normally weaned during their first winter I wouldn’t say dispersal was “long after” weaning. I suggest removing “long”.
L195 – “spacial” should be “spatial”.
L197 – quantify how many foal deaths or remove this statement.
L190-202 – I suggest removing this sentence, especially as infanticide is discussed later.
L206 – there have been reports of mountain lion predation preventing population growth in 2 feral horse populations in America, and wolf predation affected Przewalski’s horses in Mongolia.
L213-216 – there is conflicting logic here – are the mares weak because of teeth or because they have produced many foals? What evidence do you have that mares become weak and are able to forage less efficiently after producing many foals? I think it is impossible to disentangle age and teeth from effect of giving birth multiple times. I suggest removing the text about remaining in a reproductive state.
L216-217 – elaborate on how foals of older mares can become a welfare risk. Do you mean because they may be in poor condition? In a feral population are you suggesting foals be removed and bottle fed? It would be difficult to catch them!
L229 – as I said in my previous review, explain this to those who do not know about horse social structure. I suggest, “Sometimes, males of a similar age with the same father but different mothers who grew up in the same group will disperse together.”
L235 – we do not need to perpetuate patriarchal language (as I said in my previous review), especially when we know that it does not accurately represent what happens (as you said in your response). Therefore I recommend rephrasing to “… or by forming new groups with mares from pre-existing harems.”
L236 – similarly rephrase to “Shortly before becoming harem stallions, young males might …”
L270 – replace “expelled” with “recently dispersed”.
L278 – remove “or seize mares from another male,” as this is basically what “mares can also join a harem of their own volition” means in practice and is much more accurate.
L290 – change “seize” to “be joined by”.
L300 – change “the guarding” to “any guarding”.
L306 – change “against abduction” to “from joining another male”.
L327 – this citation refers to Camargue horses, not Przewalski’s.
L360 – by “expelled” do you mean “dispersed”? If so, use that word.
L374 – but could it not be due to frustration in this instance?
L421 – this is arguable. As Rubenstein and Hack demonstrated, stallions are well aware of each other’s fighting potential, but the many descriptions of multi-male bands show that they can also work together. I consider that stallions have life-long bonds, they just do not always remain in the same group. I (and others) have observed harem stallions to rush at a bachelor group, only to then play with them.
L425 – this could also be due to proximity – stallions are often on the edge of a group.
L438 – there are few studies of paternity in horses, but results are highly variable as to the number of sneak matings. I therefore suggest removing the text in parentheses also as it is contradicted by your next sentence.
L440 – change the period/full stop to a semi-colon.
L450 – the opposite could also be true as the higher rates of allogrooming result in less tension. Lansade et al. (2022) found that grooming led to more grooming.
L520 – this subheading seems unnecessary.
L546 – I don’t understand the addition of “under human supervision”. What does this mean? You have already explained what you mean by semi-feral, and feral and semi-feral horses are not under human supervision by definition.
L547 – change “own” to “associate with”.
L567 – remove ‘welfare aspects’ as a subheading. It is not at all clear what this is referring to.
L640 – “In some populations…” makes it sound like castration is a common management tool, whereas in fact it has only been used in a handful of populations. Rephrase.
L643 – harems were in fact intact, but the number of sneak matings must have been higher. Rephrase.
L647 – I think that references 117 and 113 are mis-cited.
L651 – chemical castration has been tested in feral horses too (Scully et al. 2015).
L654 – see Bechert et al. (2022) for a review of methods used in feral horses.
L682 – only infanticide was mentioned previously. Rephrase.
L710 – I think it is a bit tenuous to say you have identified welfare issues, when all are what would be experienced by wild animals and not considered welfare issues.
L730 – wild equids have been under natural selection pressures; horses have been under artificial selection for the past ~6,000 years (see my comment above).
L732 – we have selected horses through domestication that can withstand captivity, thus they have adapted to it. I think this is a spurious argument, or I am not sure what you expect stallions to have adapted to.
